# Elevated fish densities extend kilometres from oil and gas platforms

**Joshua M. Lawrence**[1]*, **Douglas C. Speirs**[2], **Michael R. Heath**[2], **Toyonobu Fujii**[3], **Finlay Burns**[4], **Paul G. Fernandes**[1]

1 The Lyell Centre, Heriot-Watt University, Edinburgh, Scotland, United Kingdom, 2 Department of Mathematics and Statistics, University of Strathclyde, Glasgow, Scotland, United Kingdom, 3 Graduate School of Agricultural Science, Tohoku University, Onagawa, Japan, 4 Marine Laboratory, Marine Scotland Science, Aberdeen, Scotland, United Kingdom

* joshua.lawrence@hw.ac.uk

**Data Availability Statement:** The data supporting the results will be archived on GitHub: https://github.com/joshua-lawrence1/2023_Lawrence_PLOS_ONE.

## Abstract

Thousands of offshore oil and gas platforms have been installed throughout the world's oceans and more structures are being installed as part of the transition to renewable energy. These structures increase the availability of ecological niches by providing hard substrate in midwater and complex 3D habitat on the seafloor. This can lead to 'hotspots' of biodiversity, or increased densities of flora and fauna, which potentially spill over into the local area. However, the distances over which these higher densities extend (the 'range of influence') can be highly variable. Fish aggregate at such structures, but the range of influence and any implications for wider fish populations, are unclear. We investigated the relationship between fish and platform areal densities using high resolution fisheries acoustic data. Data were collected in the waters surrounding the vessel exclusions zones around 16 oil and gas platforms in the North Sea, and throughout the wider area. We estimated densities of schooling fish using echo-integration, and densities of non-schooling fish using echo-counting. At 10 platforms, non-schooling fish densities were elevated near the platform relative to background levels in the equivalent wider area. The range of influence, defined here as the range to which fish densities were elevated above background, varied from 0.8 to 23 km. In areas of high platform density, fish schools were encountered more often, and non-schooling fish densities were higher, when controlling for other sources of environmental variation. This is the first time such long-range effects have been identified; previously, ranges of influence have been reported in the order of just 10s-100s of metres. These findings suggest that the environmental impact of these structures may extend further than previously thought, which may be relevant in the context of upcoming management decisions around the decommissioning of these structures.

## Introduction

The propensity of fish to aggregate around structures and objects is well known. These effects vary from brief associations with floating objects [1–3] to large scale associations with reef structures. These behaviours have been exploited commercially through Fish Aggregation

**Funding:** JML, DCS, MRH and PGF were funded by the UK Research and Innovation (UKRI) Natural Environment Research Council (NERC; https://www.ukri.org/councils/nerc/) grant number NE/T010681/1 as part of the FISHSPAMMS project in the INSITE programme. The funders had no role in study design, data collection and analysis, decision to publish, or preparation of the manuscript.

**Competing interests:** The authors have declared that no competing interests exist.

Devices (FADs) in some fishing industries [4, 5], and artificial reefs for fisheries or productivity enhancement [6–8].

Fish are also known to associate with man-made marine structures (MMS), such as oil and gas platforms [9–11]. There are currently thousands of these platforms currently installed globally, following the expansion of offshore fossil fuel exploration in the mid-20[th] century [12, 13]; the expansion of offshore renewable energy [14] will add many more MMS in the coming years. In some regions (e.g. the Gulf of Mexico), oil platforms host such consistently high fish numbers that they are targeted by both commercial and recreational fishermen, and visited by recreational SCUBA divers [15, 16]. Furthermore, a study of the oil platforms off California found them to be the most productive waters per unit area of seafloor of any area for which similar estimates exist [17].

Much of the global oil and gas infrastructure is now nearing the end of its operational life and will soon need decommissioning, meaning managers will be required to decide on the optimal approach to removing much of what was installed. 'Rigs-to-reefs' programmes, where platforms are left in place or are partially dismantled or toppled, but ultimately remain in the ocean, are in operation in several parts of the world [18, 19]. Benefits of such an approach include continuing to support the same incidental industries that utilise operational rigs (e.g. commercial and recreational fishing and other marine tourism in the Gulf of Mexico), reducing decommissioning costs [20], and lowering the risk to human life during the decommissioning process. However, in some areas, such as the North Sea, current legislation requires the complete removal of all installed structures, although derogations can be made under certain conditions on a case-by-case basis. This legislation was established following public outcry around the Brent Spar decommissioning [21]. However, there is a lack of unequivocal evidence regarding the ecological impact of these structures, and the likely consequences of their removal.

There is evidence that oil and gas platforms in the North Sea act as artificial reefs supporting a diversity of taxa and large numbers of a range of species [22–27], and potentially affecting the local ecology of those species [28]. In addition, the legally enforced 500 m safety zones surrounding North Sea oil and gas platforms allow them to act as *de facto* marine protected areas (MPAs), offering protection from certain anthropogenic pressures such as fishing and shipping. While elevated fish densities [25] and significant fish residency times [29] have been recorded around oil and gas platforms in the North Sea, there is currently little evidence to suggest that these elevated numbers extend to ranges beyond the edge of these safety zones. Such 'spillover' effects, where the benefits of an MPA extend beyond its boundaries, have been documented in other areas [30–33]. Commercial fishermen may be opposed to the abandonment of an oil platform due to the lack of access to potential fishing grounds [34], but might be able to exploit fish outside the safety zone which are present in higher densities than would be the case if the platform, and its safety zone, were removed.

Fish distributions around oil and gas platforms and other infrastructure in the North Sea have not been studied extensively. One study on a closed-down platform in the Norwegian sector found elevated fish numbers caught by gillnets in close proximity to the platform (within 150–300 m), but these dropped quickly as distances from the platform increased [25]. Another study using hydro-acoustic surveys found no increase in fish density near the same platform [35]. These studies focussed on the immediate vicinity of a single platform, not considering longer distances, or other platforms. Other work using tagged fish and trawl data in the southern North Sea similarly found a positive association between proximity to structures and fish density, although this was found to vary temporally, with species, and with structure type [36]. However, there was significant uncertainty associated with the tag-derived fish locations, and

the resolution of the trawl data and the grid used for data extraction and modelling was relatively coarse.

There remains, therefore, a need for larger-scale, but high-resolution, surveys of fish distributions around multiple platforms throughout the North Sea. Such surveys can be difficult and expensive over large areas, but fisheries acoustics provide a means to monitor fish at very fine resolution, rapidly and effectively [37]. Data collection, processing and analysis techniques are well refined and have been used extensively to study fish abundance, biomass and distribution at the ecosystem scale [38–40]. Here, we use fisheries acoustics to investigate the distributions of fish around oil and gas platforms and associated infrastructure throughout the North Sea.

Our specific aims were to analyse the relationship between fish density and oil and gas MMS for two types of fish (schooling fish and individuals) by testing three null hypotheses: i) that fish density showed no trend with distance to the nearest MMS; ii) that fish density close to MMS was not significantly higher than fish density in equivalent background areas; and iii) that fish were not associated with oil and gas MMS after accounting for other factors which may influence fish distribution.

## Materials and methods

Acoustic data were collected from the Fisheries Research Vessel *Scotia* during a North Sea trawl survey between 23$^{rd}$ July and 11$^{th}$ August 2012. A Simrad EK60 scientific echosounder collected data at 4 frequencies (18, 38, 120 and 200 kHz), transmitting at 1 Hz using a pulse duration of 1.024 ms. The 18 kHz transducer had an 11˚ beamwidth, and all other transducers had beamwidths of 7˚. Raw data were digitized and recorded as time-stamped volume backscattering strengths ($S_v$, dB re. 1 m$^{-1}$) (Fig 1) along with detected bottom depth and the vessel's GPS location for each ping. The echosounder was calibrated using standard protocols prior to the survey on 1$^{st}$ July 2012 [41].

Acoustic data were processed in Echoview [42]. Standard pre-processing steps were performed: removing bad data and the layer from 0–12 m depth (to account for the depth of the transducer and nearfield), and correcting the automatically detected seabed and adding a backstep of 0.5m to ensure no seabed was included in the analysis domain. Two data processing algorithms were then applied to the data to: 1) isolate and quantify backscatter from schools of swimbladdered fish; and 2) identify and enumerate echoes from individual fish (Single Echo Detections, SEDs).

### Fish school isolation algorithm

A multifrequency thresholding algorithm (built using Echoview's 'virtual variable' functionality) was used to identify areas of backscatter that were consistently strong across all the frequencies and indicative of schools of fish with swimbladders (such as *Clupea harengus*, herring, or *Sprattus sprattus*, sprat), as described by Fernandes [43]. In some areas, a scattering layer was present which was particularly strong at 38 kHz, but weaker at other frequencies, which is not characteristic of fish [44]. Occasionally this was strong enough to be 'accepted' by the initial algorithm as a fish school. To identify and remove the remnants of this layer, candidate schools where scattering at 38 kHz was >10dB stronger than at 200 kHz were discarded. Additionally, swimbladdered fish schools in this area generally have 'harder' edges than dense areas of the scattering layer, due to layers being horizontally extensive and continuous, by definition [45–47]. To this end, the mean $S_v$ at 38 kHz was also calculated for the area around each candidate school (using a 7 x 7 dilation filter) and any regions where the difference between

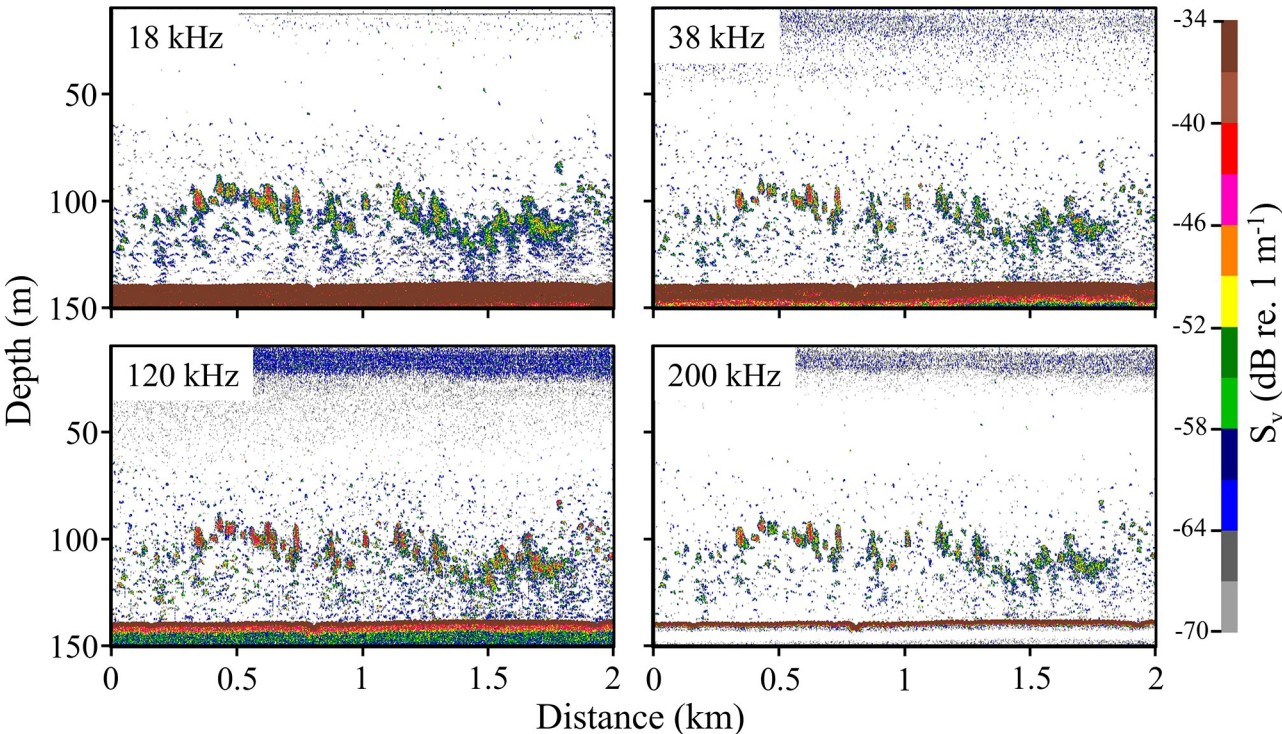

**Fig 1. An echogram collected during the North Sea trawl survey.** Acoustic data collected at 4 frequencies, 18, 38, 120 & 200 kHz, showing the seabed (region of strong scattering at ~140 m depth), fish schools (discrete regions of strong scattering across the four frequencies at ~100 m depth), and individual fish (areas of more diffuse scattering close to the seabed). These data were collected in proximity (<2 km) to one of the oil platforms surveyed.

the mean value for the candidate school and the surrounding water was less than 5 dB were discarded. This process removed echoes from the scattering layer, leaving only fish schools.

## Individual fish detection (echo-counting)

'Echo-counting', enumerating the echoes from individual fish to produce estimates of the areal density of non-schooling fish, is rarely used in the marine environment, more commonly being employed in shallow freshwater systems such as rivers [48–50]. This is, in part, due to the greater depths under consideration in marine studies, which lead to greater beam volumes, and a corresponding increase in the likelihood of encountering multiple fish in the same pulse [51]. The 'single target detection' functionality relies on the recognition of the characteristic shape and properties of the echo from a single isolated scatterer as an SED. However, it is possible for coincident echoes from multiple scatterers to be wrongly identified as a single echo (particular in regions of high backscatter). Sawada et al. [52] developed indices ($M$ and $N_v$) based on estimated local density of fish which can be used to determine areas where single targets can be detected reliably.

Here, a 50 x 5 m (horizontal x vertical) grid was used to discretise the data into subsamples for which these Sawada indices could be calculated. High density areas, identified by the thresholding algorithm (see 2.1 above) were masked from the raw 38 kHz data (i.e. setting their pixels values as 'no data') before a mean $S_v$ was calculated for each grid cell. To calculate the Sawada indices, a mean target strength (TS) was required; here, the mean TS of fish caught in the bottom-trawl survey was used. This was calculated from the catch per unit effort

(CPUE) data for the four main species of interest; a mean length was calculated for each haul in the survey, and the overall mean of these values was converted to a TS based on the TS to length equation for gadoids (TS = 20.log$_{10}$L-67.5) [37].

For each 50 x 5 m grid cell, the value of $M$ and $N_v$ were calculated, and using the thresholds suggested by Sawada et al. (1993), cells with $M < 0.7$ and $N_v < 0.04$ were considered sufficiently low density for reliable single target detection. The detected single targets were thresholded to remove small targets, below the minimum expected size of the species of interest (*Gadus morhua*, cod, *Melanogrammus aeglefinus*, haddock, *Pollachius virens*, saithe, and *Merlangius merlangus*, whiting) based on the TS of 2.5$^{th}$ percentile of the length distribution of the trawl survey data (-50.47 dB).

## Data export and other sources

Several datasets were exported from Echoview: 1) the fish school $S_v$ echogram was integrated over 50 m segments of transect, or elementary distance sampling units (EDSUs), to give values of the Nautical Area Scattering Coefficient (NASC); 2) the SEDs falling within the areas retained by the 'Sawada' filter (described above) were exported individually, including details of their TS, depth and time/date stamp; 3) the seabed depth was exported for each ping in the survey.

The SEDs were assigned to the EDSU in which they were detected based on the time/date stamp of their record. Survey effort varied between EDSUs because of variation in ping volume due to changes in seabed depth and the variable number of pings per EDSU due to changes in vessel speed. To account for this, the total volume of water insonified in each EDSU (including the 'double counting' of overlapping pings), and the corresponding equivalent area of sea-surface was calculated. The height of the cone ($H_{equ}$) with the same volume as the sampled portion of the ping (from 12 m depth, following exclusion of the near-field, to the seabed depth ($D_{ping}$) for that ping) was calculated as:

$$H_{equ} = \sqrt[3]{D_{ping}^3 - 12^3} \tag{1}$$

The volume of each ping was then calculated as:

$$V_{ping} = \frac{\pi}{3} . \tan\frac{\alpha}{2} . \tan\frac{\beta}{2} . \sqrt[3]{H_{equ}} \tag{2}$$

where $\alpha$ and $\beta$ are the major- and minor-axis beam angles, respectively. Areal fish density (m$^{-2}$) in each EDSU was then estimated as:

$$Dens_{EDSU} = \frac{SEDs_{EDSU}}{n_{EDSU}} . \sum_{i=1}^{n_{EDSU}} \frac{H_{equ_i}}{V_{ping_i}} \tag{3}$$

where $SEDs_{EDSU}$ is the total number of SEDs recorded in the EDSU and $n_{EDSU}$ is the number of pings in the EDSU. This is equivalent to the methods of Kieser and Mulligan [53], except calculating fish density per unit area, instead of per unit volume.

Data on the locations of oil and gas platforms were extracted from the 2013 OSPAR inventory of offshore infrastructure (OSPAR, 2013) [54]. A density surface of platforms (Fig 2a) was created in ArcGIS (using the 'kernel density' function), to be used as a better measure of the local environment than using simple proximity to the nearest platform, due to their highly variable grouping.

The locations of oil and gas pipelines throughout the region were obtained from the European Marine Observation and Data Network (EMODnet). These were filtered to remove those

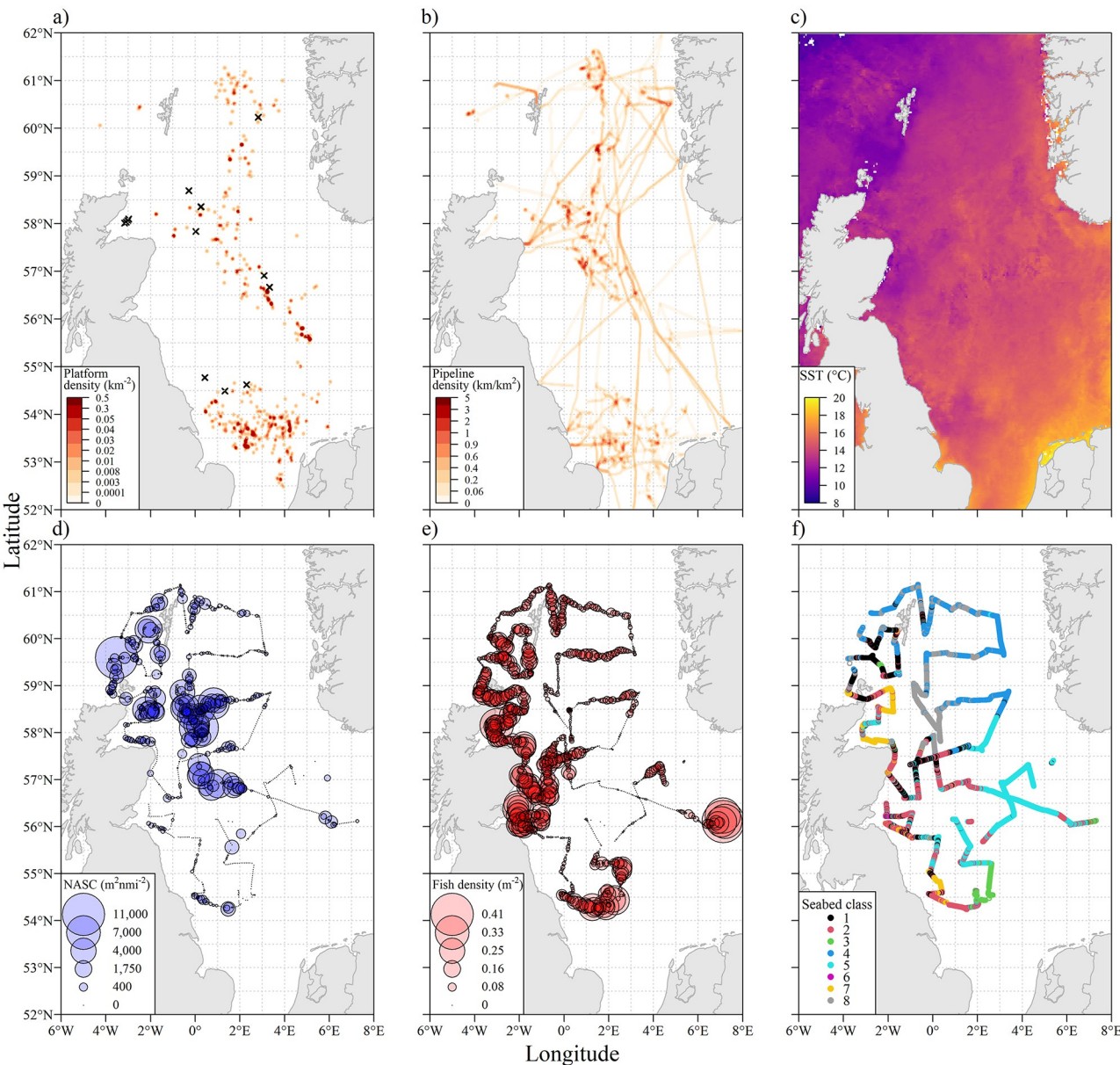

**Fig 2. Maps of fish distributions and environmental data.** Maps of the North Sea showing: a) oil and gas platform areal densities, and the locations of platforms surveyed to within a 1 km range (black crosses); b) oil and gas pipeline areal densities; c) Aqua-MODIS satellited derived Sea Surface Temperature (˚C) from August 2012; d) fish school density where the area of each circle is proportional to the Nautical Area Scattering Coefficient (m2nmi-2); e) density of non-schooling fish where the area of each circle is proportional to numbers per m2; and f) acoustically derived seabed habitat class (labelled 1–8) overlaid on the ship's cruise track.

with a listed installation date of 2013 or later, and a similar density surface (Fig 2b) to those created for platforms (using the 'kernel density' functionality in ArcGIS) was produced.

Sea surface temperature (SST) data were obtained from NASA's OceanColor web portal, as 4 x 4 km resolution netCDF files of Aqua-MODIS satellite data (Fig 2c). For each 50 m EDSU, mean monthly temperatures for July and August were extracted, and a mean of those values was used in the subsequent analysis since the survey period spanned the end of July/start of

August. Near-seabed water temperatures were obtained from measurements taken throughout the survey using a temperature profiler. These data were obtained from the ICES ocean hydro-chemistry data portal (https://data.ices.dk/). Each EDSU was assigned the temperature value from the maximum depth of the most proximate temperature profile obtained.

Due to potential diel variation in fish behaviour (e.g. schools dispersing and demersal fish moving off the seabed at night) solar elevation was calculated for each EDSU using the package *suncalc* in R, and represented as a binary variable of day/night signifying the sun being above (elevation $>0$) or below ($<0$) the horizon, respectively.

Echoview's 'habitat classification module' was used to perform unsupervised clustering of acoustic transmissions (into unnamed seabed 'classes') based on acoustically-derived seabed characteristics [55]. The characteristics used were roughness, hardness, depth, kurtosis, skewness, and length, rise time, and maximum $S_v$ of the bottom echo. (For additional details, see the Echoview online help for 'Bottom Classification'). These features were extracted for every $10^{th}$ ping, and the extracted pings were grouped into arbitrary seabed classes by the clustering algorithm. In order to avoid unrepresentative seabed classes being assigned to short (in duration) EDSUs which might only include a single clustered ping, the modal value of seabed class for clustered pings within the period from one minute before the start to one minute after the end of the EDSU was assigned as the seabed class for the EDSU.

## Data close to structures

Data recorded near platforms were examined on a platform-by-platform basis. 16 platforms were approached to within 1 km by the survey vessel (Table 1), although data were collected to a minimum distance of 500 m at 15/16 of these, due to the 500 m safety zones. For each platform, EDSUs were identified to which that platform was the most proximate, up to a maximum distance of 10 km from the platform. Additionally, for each platform, a 'baseline' fish density dataset was identified. These were the data which were within 10 m depth of the mean of the data near the platform, within 1° latitude of the platform location, but >25 km from any platform. These data were subset to match the proportional distribution of the near-platform

**Table 1. Details of platforms the survey vessel approached to within 1km.**

| Platform | Water depth (m) | Platform type | Status | Product | Substructure weight (t) | Production start |
|---|---|---|---|---|---|---|
| a | 45 | Fixed steel | Operational | Oil | 1,976 | 1981 |
| b | 101 | Concrete | Derogation | Gas | 386,000 | 1977 |
| c | 45 | Fixed steel | Operational | Oil | 3,225 | 1981 |
| d | 109 | Fixed steel | Operational | Oil | 5,983 | 2000 |
| e | 45 | Fixed steel | Operational | Oil | 1,537 | 1981 |
| f | 146 | Fixed steel | Operational | Oil | 22,555 | 1976 |
| g | 144 | Fixed steel | Decommissioned | Oil | 14,300 | 1976 |
| h | 54 | Fixed steel | Operational | Gas | 1,152 | 2006 |
| i | 66 | Fixed steel | Operational | Oil | 13,184 | 1990 |
| j | 40 | Fixed steel | Operational | Oil | 950 | 2009 |
| k | 117 | Floating steel | Operational | Oil | 0 | 1981 |
| l | 70 | Fixed steel | Operational | Oil | 583 | 1977 |
| m | 28 | Fixed steel | Operational | Gas | 550 | 2005 |
| n | 62 | Fixed steel | Operational | Gas | 2,300 | 2013 |
| o | 45 | Fixed steel | Operational | Oil | 730 | 1981 |
| p | 70 | Fixed steel | Operational | Oil | 5,275 | 1978 |

data across seabed classes. To compare near-platform data with this 'baseline', the near-platform data were grouped into 500 m distance bins (0–500 m, 500–1000 m, 1000–1500 m etc., from the platform), and these were compared to the 'baseline' using a Welch test [56] (a modified t-test suitable for comparing datasets with unequal variances).

Table 1; 'Platform' is an arbitrary identifier, and specifies the corresponding subplot in Fig 3. Note, the substructure weight of 0 for platform *k* is indicative of it being a floating platform, and while the production start date of platform n is after the survey was conducted, the platform was installed in 2011. Details were obtained from OSPAR's 2013 Inventory of Offshore Installations.

For each platform dataset, a linear model of individual fish density against distance from platform was fitted. This linear model was used to estimate the horizontal range of influence (HRI) for each platform, which we define similarly to Stanley and Wilson's [57, 58] 'area of influence', as the distance to which fish densities are higher than at greater distances.

The mean HRI of platforms was calculated as the mean distance at which modelled fish density equalled the median of their respective baselines (for platforms where the linear model had a negative slope). To ensure this distance was not influenced by the subsampling for the baselines (to match the proportional distribution of the platform data across seabed habitat classes), the subsampling was repeated 30 times, and an overall mean distance was calculated.

## Non-linear modelling of fish density with explanatory variables

The relationships between recorded fish densities and the various explanatory variables were investigated using generalised additive modelling (GAM) and generalised additive mixed modelling (GAMM), implemented in the *mgcv* package for R. Due to the extreme zero inflation and skew of the fish school acoustic density (NASC) data, a two-stage modelling process was used. First, school presence/absence was modelled using a binomial error distribution and then log-transformed fish school NASC, where >0, was modelled using a Gaussian error distribution. Individual fish (SEDs) were modelled as counts per EDSU using a negative-binomial error distribution, and using the total equivalent sampled area of the pings in the EDSU as an offset. Continuous explanatory variables were included in the models as smooth terms (thin-plate regression splines), limited to 6 basis dimensions to avoid over-fitting and ensure the biological interpretability of the resulting curves. For models of fish school presence/absence and density, SST was used, whereas for models of non-schooling fish, sea bottom temperature was used. Day/night, acoustically derived seabed 'class', and the type [59] of the nearest oil and gas platform to the EDSU mid-point (floating steel, fixed steel or gravity-based concrete) were included in models as factor variables. Only data within 25 km of oil and gas platforms were included in the non-linear modelling analyses.

Three models (fish school presence/absence, fish school NASC where >0, and SED density) were calculated; to avoid a type I error when fitting multiple models, a threshold (alpha) of 0.01 was used to determine significance (as opposed to the more standard 0.05) of model terms.

## Results

Over 5,000 km of active acoustic data collection was completed, which was broken down into 106,020 50 m EDSUs. Acoustic densities of schooling fish were recorded up to a maximum NASC (in a single EDSU) of 526,262 $m^2nmi^{-2}$ (Fig 2d), and the maximum number of individual fish detected in an EDSU was 781 (Fig 2e). Acoustic characterisation of the seabed using Echoview's 'habitat classification' module classified the seabed into 8 distinct classes (Fig 2f) based on the extracted acoustic properties.

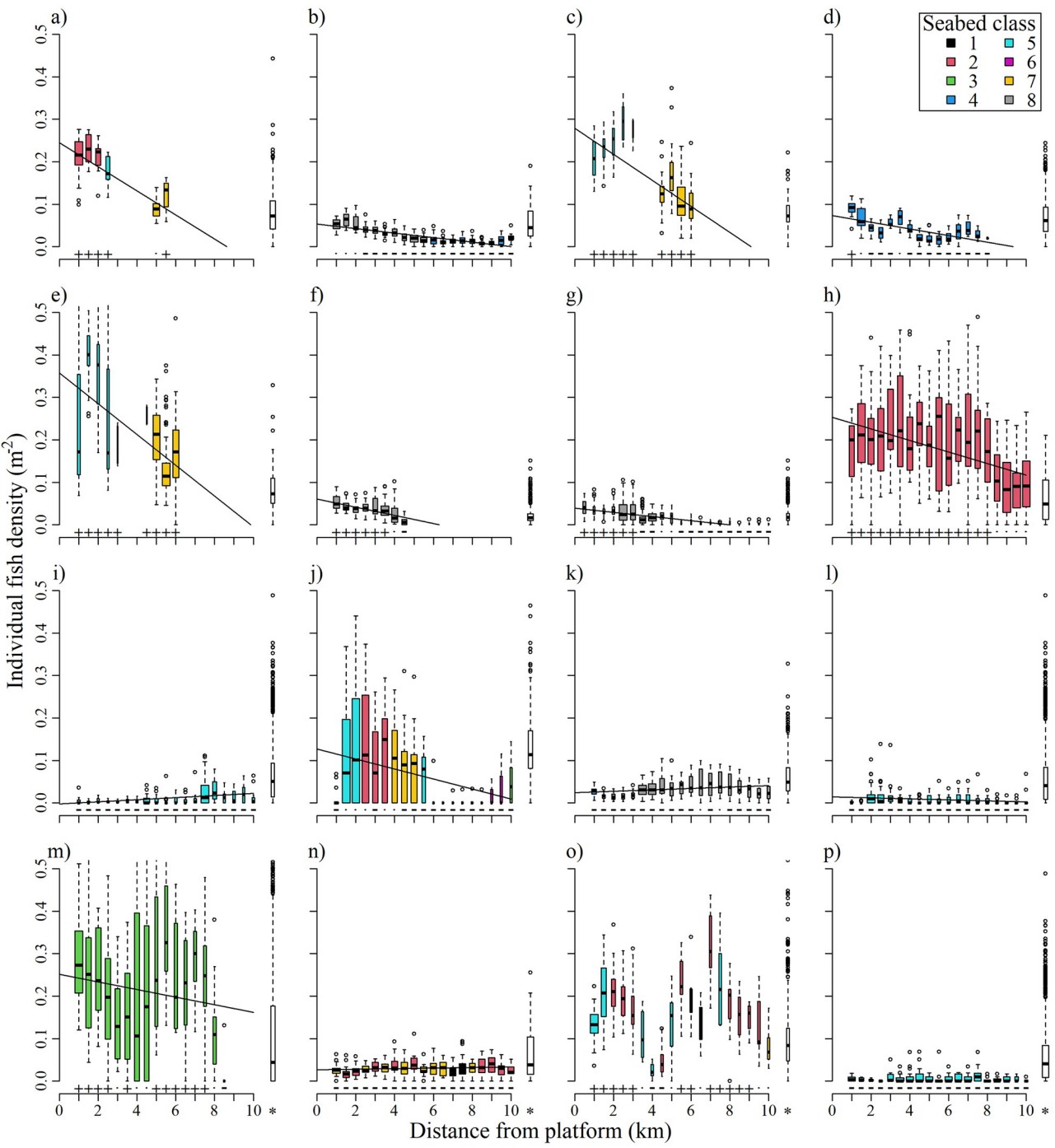

**Fig 3. Boxplots of densities of individual fish (i.e. non-schooling) vs distance from platform.** Each subplot shows data closest to each of the 16 platforms surveyed at ranges <1 km. Data are binned to 500 m intervals; the x-axis values are the upper limit of bins (i.e. the box plotted at 1 km represents the data in 500–1000 m bin). Box colour represents the modal seabed class of the data for that box (see legend). Black lines are the lines of best fit of linear models, shown only where the relationship was significant (i.e. $p < 0.05$). Plots are ordered by the model fit ($R^2$ value). Boxes to the right of each subplot (marked * on x-axis) represent the equivalent 'baseline' fish density data; these are >25 km from any platform, from water depths within 10 m of the mean for each subplot, with the proportional distribution across seabed classes, and within 1° latitude of the mean of the data. The result of Welch tests for differences between the data in each box and the relevant 'baseline' are shown above the x-axis (with + and—signifying a significantly higher or lower mean than the baseline, respectively). Details of each platform (including water depth and platform type) are given in Table 1.

## Data close to structures

The data at each platform that was visited to within a 1 km range (details in Table 1) revealed a large degree of inter-platform variability in the gradient of individual fish density (Fig 3). At most platforms, a significant negative relationship was found between individual fish density and distance from platform. This simple analytic approach was not possible with the fish school NASC data, due to the extreme zero-inflation and skew of data. At 10 of the 16 platforms, fish densities close to the platform were elevated above the baseline level. At these platforms, the mean HRI was 7.2 km with a range from 0.8 to 23.2 km.

## Relationship between fish density, platform density and other variables

Of the 106,020 total 50 m EDSUs, ~36,000 were within 25 km of oil and gas platforms; these data were used to build models to investigate the relationships between fish densities and the potential explanatory variables of interest. Based on the mean HRI calculated from the near platform data, the density kernels of platform and pipeline density were produced using a 'search radius' (the range at which a single platform or pipe's influence drops to zero) of 7.2 km.

The GAM modelling fish school presence/absence exhibited a degree of spatial autocorrelation in the residuals. Visual inspection of the variogram showed the range of this autocorrelation to be ~1500 m (S1 Fig); to avoid the underestimation of errors around coefficient estimates, the data were aggregated to a support of 1500 m (i.e. the data from adjacent 50 m EDSUs were averaged, either as a modal or mean value, in sets of up to 30) and the model was re-fitted. In the aggregated model, the likelihood of fish schools being encountered was found to be higher with increasing platform density, in areas of increased water depth and increased SST (Fig 4a–4c), and in daylight hours (S1 Table). No relationship was found with pipeline density, and the only difference identified between platform types was that fish schools were less likely to be detected in EDSUs closest to floating platforms than in those closest to the other two types (fixed steel or gravity-based concrete) (S1 Table).

Spatial autocorrelation was also identified in the residuals of the GAM modelling the acoustic density of fish in schools (where present) (S1 Fig). Because the fish density data was continuous, a GAMM was used, and was fitted to include a spatial correlation structure (using a spherical model with a nugget effect, deemed appropriate from inspection of the variogram). The only significant effect identified by this model was that school density was found to be higher in the daytime than at night (S2 Table). It was notable that this model fitted the data relatively poorly (explaining just 3% of the variation in the data).

The GAM modelling individual (non-schooling) fish density also displayed spatial autocorrelation in the residuals (S1 Fig), with a similar structure to that of the fish school presence/absence model, so the model was re-fitted using the same 1500 m aggregated data. This model showed that the density of non-schooling fish increased with increasing platform density, was higher in deeper areas and of intermediate bottom water temperature (Fig 4d–4f), and during daylight hours (S3 Table). No relationship was found with pipeline density, or with platform type.

The fish school presence/absence and the non-schooling fish models also showed some significant effects between seabed habitat classes (S1 & S3 Tables), however these are difficult to interpret as the classes are not defined beyond their arbitrary numeric labels. These significant relationships do, however, reinforce the value of the inclusion of seabed habitat in the models (and so controlling for differences between the seabed type).

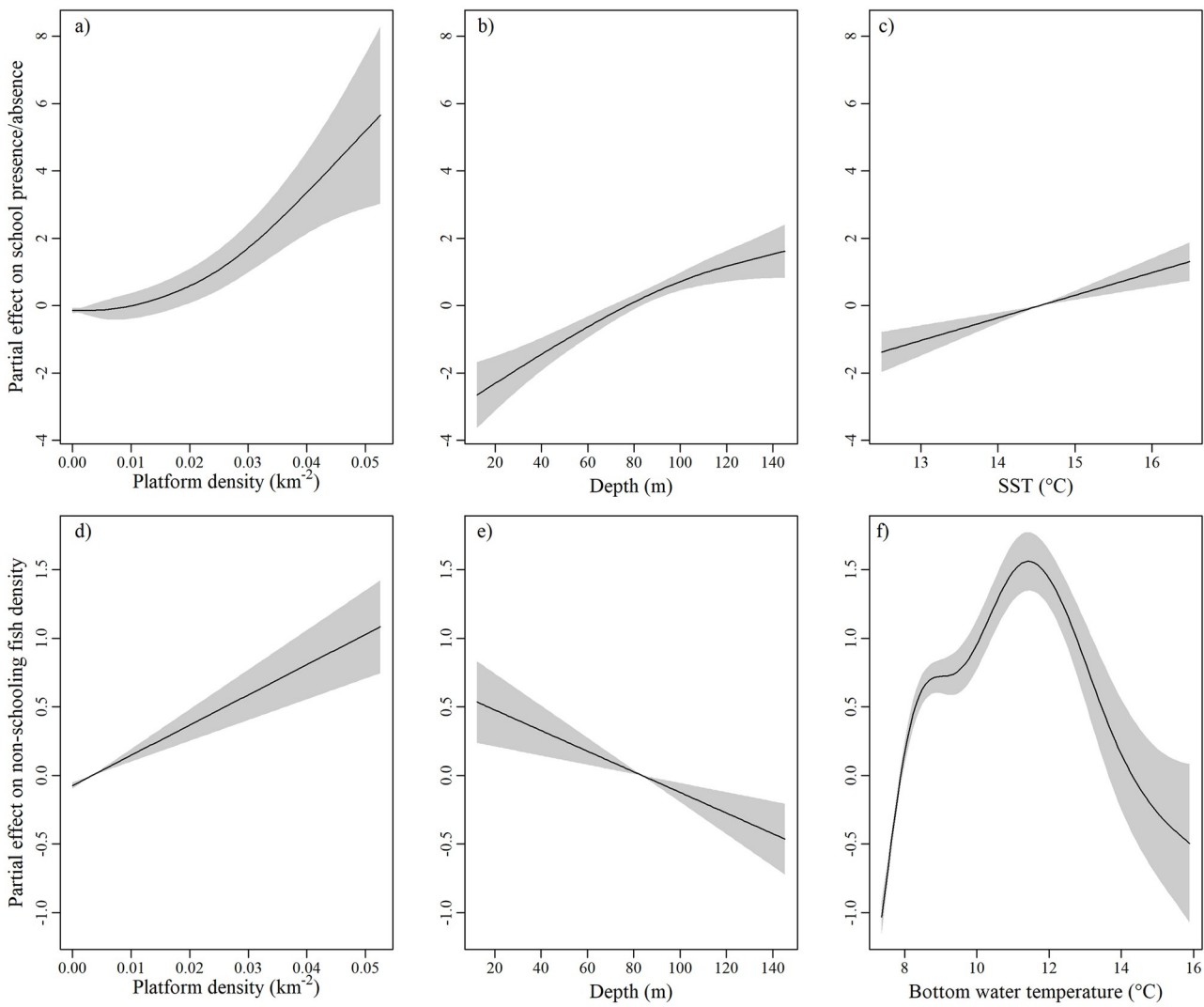

**Fig 4. Modelled relationships between fish density and explanatory variables.** Plots showing the shape of fitted explanatory terms found to be significant (at α = 0.01) during the generalised additive modelling process. Boxes a-c are terms from the model of fish school presence/absence, showing the relationship with a) platform density, b) depth and c) sea surface temperature (SST), and d-f are terms from the model of non-schooling fish density, showing the relationship with d) platform density, e) depth and f) bottom water temperature.

## Discussion

We found strong evidence that fish, both schooling and non-schooling, associate with oil and gas platforms in the North Sea. The densities of non-schooling fish were found to be higher, and fish schools were more likely to be encountered, in areas of high platform density. Furthermore, the evidence suggests that these associations exist over long ranges, up to the order of several kilometres, extending well beyond the 500 m safety zones in place around North Sea platforms. Previous work in the North Sea and elsewhere found similar elevated densities of fish in close proximity to platforms [25, 60], but reported that fish numbers dropped rapidly as distance from the platform increased beyond just 10s or 100s of metres. Here however, the horizontal range of influence ranged between 0.8 and 23.2 km (although it must be noted that

most of the data was collected >500 m from a platform). This wide range emphasises how variable this relationship is between individual platforms.

This new evidence is timely and relevant in the context of the upcoming challenges associated with the decommissioning of a large portion of the North Sea's oil and gas infrastructure as it reaches the end of its operational lifespan. Of particular importance is the evidence that the range of influence of many of these structures is significantly greater than their immediate physical footprint, and even the safety zones in place around them. While current legislative decommissioning requirements for platforms necessitate complete removal, there is a lack of evidence regarding the ecological impacts these structures, and their potential removal, may have on the ecosystem [61, 62]. Evidence that fish associate with these structures, especially across such large scales, highlights the possibility that there may be measurable benefit provided by these structures, and that their complete removal may actually be detrimental to the environment [62] and to the interests of relevant stakeholders. However, further work is needed to investigate the mechanistic causes of the relationships found in this work. Importantly, it must be considered whether there is evidence of enhanced productivity at these structures [6, 17], or if the observed trends in fish density with distance (and the elevated fish densities in close proximity to the structures) are simply due to the attraction and aggregation of the local fish population. Either way, the evidence presented here suggests that fish associating with these platforms are not always confined to the immediate proximity of the structure themselves.

That pipeline density appears to have little effect on the density of both schooling and non-schooling is somewhat contrary to findings from some other studies, but may be due to the different survey techniques used. Work using video surveys from remotely operated vehicles [63] and visual surveys from manned submersibles [10] have found increased fish densities around pipelines compared to the surrounding seafloor. However, in particular they noted that it was pipeline spans (i.e. pipeline sections elevated off the seabed), or where a pipeline was undercut, where the highest densities of fish were found. These fish would be unavailable for detection by the methods used in our study. Fisheries acoustics deployed from a vessel at the surface can only detect fish to which the echosounder has a clear line of 'sight'. Furthermore, due to the curvature of the leading edge of the acoustic beam, an acoustic 'dead-zone' exists where objects in very close proximity to the seabed or other hard structure, cannot be detected [64, 65]. These factors in combination mean that elevated densities of fish in very close proximity to pipelines would not be detected, if present.

There are some considerations which limit the extent to which further conclusions can be drawn from this work. Importantly, with the available data, the detected fish could only be divided into 'schooling' and 'non-schooling' fish, rather than by species. In order to attribute acoustic backscatter or counts of single echo detections to species, precise proportions of each species present and accurate length distributions are needed, but were unavailable for this work. In particular, for the individual fish, treating all fish as a single group will make interpretation of some results difficult. The relationship with sea-bottom temperature will, for example, be a compound relationship of the species-specific relationships which might exist, and have been documented elsewhere [66–68]. Although information on species composition was available from the accompanying bottom-trawl survey, the trawl samples were not taken close to the oil and gas platforms. However, further inspection of the two datasets may provide some evidence for species composition, particularly when combined with detected fish lengths.

Additionally, the use of data on fish sizes (during the filtering of the SEDs throughout the water column) obtained with a bottom-trawl may bias the calculation of mean target strengths due to the non-random vertical distribution of different size classes of the species of interest.

Juvenile gadoids are known to spend a larger proportion of the time in the water column than larger adults [69, 70], and so the length data used may be biased towards larger individuals. It is possible that this caused the underestimation of local fish density, and the subsequent inclusion of grid cells which erroneously passed through the filtering process [52]. Here, the bottom trawl data was the only alternative evidence for fish sizes available, and its use was necessary to avoid circularity in the filtering process [52]; ideally, pelagic trawling would have provided data on the size distribution of fish in the water column to complement the data collected with the bottom-trawl. Future work seeking to use the combination of echo-integration and echo-counting applied here, should set out to collect data on all fish 'groups' recorded; schooling fish, fish on or near the seabed, and individual fish found in midwater.

Further work could also consider the vertical distributions of fish in the context of proximity to oil and gas platforms. Several gadoid species are found in midwater as well as near the seabed, but oil platforms provide hard substrate throughout the vertical extent of the water column, and may thus alter the vertical distributions of nearby fish, although evidence for this may be limited. For example, saithe have a semi-demersal distribution [71], frequently foraging in midwater, and have been recorded throughout the water column at a North Sea oil platform [27]. In the same study, however, cod were only found in the deepest depth-stratum sampled, suggesting the oil platform may not cause predominantly demersal fish to move higher into the water column. It is noteworthy though, that the study used a relatively large mesh size for sampling, and so small fish, and any changes to their vertical distribution, may have gone undetected. Elsewhere, a study of fish production at oil rigs of the coast of California found that average production per unit area seafloor at the base of platforms (the bottom 2 m vertically) was more than double that of the midwater portion, despite the much greater extent of the latter [17], suggesting the majority of fish remain bottom-associated, even in the presence of the vertical platform structure. Despite these studies, it would nonetheless be valuable to investigate the impacts of oil platforms on the vertical distribution of fish in more detail, particularly to consider these effects over a broader horizontal scale than has been previously, in light of the long-range influence of oil platforms on fish densities reported here.

The differences in fish densities associated with each seabed habitat are also difficult to interpret. The unsupervised clustering performed by Echoview's 'habitat classification' module groups data based on the acoustic properties of the seabed, but makes no inference or assertion about what seabed type is represented by each assigned class. While some acoustic seabed properties are easily interpretable (e.g. seabed hardness or roughness), the classification is highly multi-dimensional, and the separate classes cannot be defined from the data available. More detailed work on this, involving drop-camera and/or grab sampling, and reconciliation with existing habitat maps [72], would allow the assigned seabed classes to be ground-truthed, defined, and validated.

More work is also needed to fully understand the causes of the inter-platform variability in the trends of fish density and distance from platform. While the modelling results presented here demonstrate a general trend of fish density increasing with the local density of platforms, the inspection of the platform-by-platform data demonstrates strong variability. Differences in platform design and size, and the habitat (depth, substrate, and hydrodynamic conditions) in which a platform is located may determine the influence any given platform has on the local fish population. Focussed studies of fish around platforms across the range of these variables will allow the determination of those factors which control the relationships described here.

It is also important to understand the influence of man-made structures on fish migratory and spawning behaviour. The data used in this work were collected in late summer, outside of the periods of either migration or spawning in any of the most likely species to have made up the demersal fish assemblage encountered [73]. However, the data collection period did

overlap with the start of the autumn spawning period of North Sea herring, likely the main constituent of the fish schools detected here, which may have affected the strength of associations observed. Herring are demersal spawners, returning repeatedly to traditional spawning grounds around the Scottish coast. These broad areas have relatively low overlap with those areas of high densities of oil and gas platforms, which are generally further offshore in the northern North Sea [59, 74]; as such, the associations between fish schools and oil platforms observed here may in fact be stronger at other times of year when fish are not associating with, or moving towards, spawning grounds in other areas.

The effects of increased availability of hard substrate (due to the presence of oil platforms), as well as locally increased densities of fish, may also have an effect on spawning or migration behaviour. These effects may be particularly important for those species, e.g. saithe [73], with distinct spawning areas which overlap with, or are in close proximity to, areas with high densities of oil and gas infrastructure. To examine these possible effects, as well as to support the findings of this work, similar studies with data collected at other times of year, particularly during spawning or migration of key demersal species (generally the first two quarters of the year), as well as investigations into evidence of spawning at these sites, would be invaluable.

An additional factor to be considered when investigating the influence of structures on fish densities and distributions is fish size, and whether the horizontal trends in fish density reported here are consistent across fish sizes, or if, for example, larger fish are more likely to associate with platforms than smaller fish. Target strength data (when combined with appropriate target strength-length relationships, the selection of which is ideally informed by data from trawling) provide information about fish size, and so could be used to examine any trends in fish size relating to proximity to oil platforms. These trends may in turn vary with seasonal changes in spawning and migratory behaviour, which are themselves age/size dependant. More work is needed to investigate these potential effects, and better understand the influence platforms have on the life history of local fish populations.

Studies of spawning activity at oil platforms, and a better understanding of the demography of fish associating with platforms, would also contribute to addressing the key area of uncertainty remaining for managers who must decide the fate of these oil and gas platforms. This uncertainty is around whether the increased numbers of fish recorded at and in the vicinity of platforms is due to increased local production, or simply due to aggregation of the surrounding fish population [6, 75–77]. Elsewhere, similar platforms have been found to have the highest secondary production per unit area of seafloor of any measured system [17]; the data presented here, and other datasets currently being collected, will allow progress to be made towards answering this key question.

## Conclusions

The work described here provides evidence that both schooling and non-schooling fish associate with oil and gas platforms in the North Sea over long distances. Where previously fish densities were only thought to be elevated in very close proximity (less than a few 100s of metres) to platforms, here they were often found to remain above baseline levels for several kilometres, well beyond the 500 m safety zones in place around active platforms. Uncertainty remains around whether these trends are due to aggregation of fish or increased local production, and around the causes of the inter-platform variability seen in the observed trends, but this work presents the first suggestion that the ecological impact of these structures, particularly on fish populations, may be wider ranging that previously thought.

## Supporting information

**S1 Fig. Residual variograms.** Omnidirectional variograms of the residuals of the models fit for fish school presence/absence (a & b), fish school density (c &d) and individual fish (SED) density (e & f), before (a, c & e) and after (b, d & f) the implantation of data aggregation or correlation structure in the model.
(DOCX)

**S1 Table. Modelling results for factor variables in the model of fish school presence/absence.** Modelling results for factor variables from the GAM modelling fish school presence/absence, showing term estimates, standard errors, z- and p-values. The omitted factor levels (Platform Category: Fixed, 'night', 'Bottom class 1') are constituents of the model intercept. GBC abbreviates gravity-based concrete.
(DOCX)

**S2 Table. Modelling results for factor variables in the model of fish school density, where present.** Modelling results for factor variables from the GAM modelling fish school density, where present, showing term estimates, standard errors, t- and p-values. The omitted factor levels (Platform Category: Fixed, 'night', 'Bottom class 1') are constituents of the model intercept. GBC abbreviates gravity-based concrete.
(DOCX)

**S3 Table. Modelling results for factor variables in the model of individual fish density.** Modelling results for factor variables from the GAM modelling individual fish (SED) density, showing term estimates, standard errors, z- and p-values. The omitted factor levels (Platform Category: Fixed, 'night', 'Bottom class 1') are constituents of the model intercept. GBC abbreviates gravity-based concrete.
(DOCX)

## Acknowledgments

The authors thank the captain and crew of *FRV* Scotia for their work during the collection of the data. We would like to thank NASA's OBPG & OB.DAAC for the free access to the sea surface temperature data used here. We are also grateful to the two anonymous reviewers for their comments and feedback on our initial submission.

## Author Contributions

**Conceptualization:** Douglas C. Speirs, Michael R. Heath, Toyonobu Fujii, Finlay Burns, Paul G. Fernandes.

**Data curation:** Joshua M. Lawrence, Toyonobu Fujii, Finlay Burns.

**Formal analysis:** Joshua M. Lawrence.

**Funding acquisition:** Douglas C. Speirs, Michael R. Heath, Paul G. Fernandes.

**Investigation:** Toyonobu Fujii, Finlay Burns.

**Methodology:** Joshua M. Lawrence, Douglas C. Speirs, Michael R. Heath, Toyonobu Fujii, Finlay Burns, Paul G. Fernandes.

**Project administration:** Joshua M. Lawrence, Paul G. Fernandes.

**Resources:** Finlay Burns, Paul G. Fernandes.

**Software:** Joshua M. Lawrence.

**Supervision:** Paul G. Fernandes.

**Visualization:** Joshua M. Lawrence.

**Writing – original draft:** Joshua M. Lawrence.

**Writing – review & editing:** Joshua M. Lawrence, Douglas C. Speirs, Michael R. Heath, Paul G. Fernandes.

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
