## [Decision Letter · Decision Letter 0]

31 Jan 2024

PONE-D-23-35757Elevated fish densities extend kilometres from oil and gas platformsPLOS ONE

Dear Dr. Lawrence,

Thank you for submitting your manuscript to PLOS ONE. After careful consideration, we feel that it has merit but does not fully meet PLOS ONE’s publication criteria as it currently stands. Therefore, we invite you to submit a revised version of the manuscript that addresses the points raised during the review process. The manuscript presents a clear analysis of how fish densities vary with distance from oil platforms and pipelines.  There are important findings which will stimulate much interest and new studies.  Overall the manuscript is organized and clearly presented.  The reviewers have some comments, which are mostly of a technical/clarifying nature and should be relatively straightforward to accommodate.  Please adjust according to these comments.  I agree with reviewer 1 that the Figure S1 and Table S1 could be included in the main manuscript.

We look forward to receiving your revised manuscript.

Kind regards,

Brian R. MacKenzie, Ph. D.

Academic Editor

PLOS ONE

Journal Requirements:

3. Thank you for stating the following in the Acknowledgments Section of your manuscript: "The analysis and write up were funded by the UK Natural Environment Research Council (NERC) as part of the FISHSPAMMS project (grant number NE/T010681/1) under the INSITE programme."

Please remove any funding-related text from the manuscript and let us know how you would like to update your Funding Statement. Currently, your Funding Statement reads as follows: "JML, DCS, MRH and PGF were funded by the UK Research and Innovation (UKRI) Natural Environment Research Council (NERC; https://www.ukri.org/councils/nerc/) grant number NE/T010681/1 as part of the FISHSPAMMS project in the INSITE programme. The funders had no role in study design, data collection and analysis, decision to publish, or preparation of the manuscript."

Reviewers' comments:

Reviewer's Responses to Questions

**Comments to the Author**

1. Is the manuscript technically sound, and do the data support the conclusions?

Reviewer #1: Yes

Reviewer #2: Yes

2. Has the statistical analysis been performed appropriately and rigorously? 

Reviewer #1: Yes

Reviewer #2: Yes

3. Have the authors made all data underlying the findings in their manuscript fully available?

Reviewer #1: Yes

Reviewer #2: Yes

4. Is the manuscript presented in an intelligible fashion and written in standard English?

Reviewer #1: Yes

Reviewer #2: Yes

5. Review Comments to the Author

Reviewer #1: Line 158: corrected depth please clarify

Line 187: please name the ICES data portal

Page 13: I would prefer to include the Figure and Table from the Supplementary material here in the main MS

Line 283: visual inspection of the variograms or formal test ?

Line 333: what about the dead zone at the surface? Was a drop keel used?

Line 355: Sediment maps for the North Sea should be available from geological surveys or projects

Page 18: Please mention that the results and conclusions may warrant a validity check with an additional data set, e.g. from quarter 1 survey or another year or a special study with bottom and pelagic trawl catches in addition to the acoustics

Reviewer #2: Overall a nice paper. The authors have done considerable work to address fish distribution and proximity to platforms. I think the manuscript could be strengthened by the addition of a couple of analyses that should not be too arduous to complete. I provide more detail in general comment #4 and specific comment #22. My recommendation is major revisions.

General Comments:

Lines 17-19. I find this statement interesting. Is the assumption that the authors are referring to the pelagic environment, rather than the benthic and/or demersal environments? If so, I don’t quite agree that the pelagic is featureless (and I don’t understand the use of “relatively”. Relative to what?). I think there are features, but we as humans either don’t understand them and consequently don’t measure them, or choose to ignore them. If the benthic habitat is to be considered, then I really don’t agree with it being featureless. I think this statement needs more detail and clarification.

An echogram would be very useful to give perspective and context.

I wasn’t sure how day/night fit into the analyses. The results state day/night differences, but were the data separated into day & night prior to applying statistics or was day/night a covariate? For example, Figure S1 shows only one trend line per graph, suggesting the data were pooled day & night, but if there were day/night differences, shouldn’t there be two graphs per test?

Was it possible to calculate fish length from the TS measurements? It would be very interesting to see if there were any relationships between fish length and proximity to platforms.

Specific Comments:

Line 20. There is still considerable debate about “spill over”. I would temper this statement with “potentially”.

Line 79. Delete “very”.

Line 90. Please define what “the region” refers to. The authors have referred to a number of different regions around the globe. Maybe rewrite to “... platforms throughout the North Sea.” if they are only referring to the North Sea.

Line 106. Please provide the beam widths. This is important information for some of the processing steps.

Line 110. Should the comma after “performed” be a colon?

Line 112. Was there a backstep applied to the seabed echo detection?

Lines 123-124. Is there a reference for “harder edges”?

Lines 124-126. Please state how the “area around each candidate school” was selected.

Line 139. Please state which dimension each value refers to, e.g., 50 m horizontal by 5 m vertical.

Line 141. Please define “masked”.

Line 143. What type of trawl was used? If a bottom trawl, do the authors suspect a bias in the TS calculations for pelagic scattering? For example, juvenile gadoids are often in the water column, whereas adults are demersal. Using a TS for adults would not be representative of the individuals in the water column.

Lines 166-168. I recommend the authors look at Kieser and Mulligan, 1984, Analysis of echo counting data: A model, Canadian Journal of Fisheries and Aquatic Sciences, 41: 451-458 as a method to estimate echo count density.

Line 187. A URL to the ICES data portal would be useful.

Line 191. What were the criteria for day and night from sun elevation?

Line 204. “Data” is plural. Replace “was” with “were”.

Line 229. “Gaussian” should be capitalized.

Line 238. Specify which analyses. In the previous section, data >25 km were used for the baseline.

Table 1 vs. Figure S1. I like the figure over the table. I feel the figure is more informative than the table, so I suggest replacing Table 1 with Figure S1.

Paragraph beginning on line 270. I would like to see a figure representing the spatial autocorrelation. This could be in the supporting documentation.

Line 292. The validity of this paragraph is impossible to evaluate without any statistical test, results, or figures. Either provide details or delete.

The discussion should include a paragraph on seasonality and migration behaviour. It would be interesting to know how their survey timing fits into the overall distribution of the key species.

The discussion should also cover pelagic vs. demersal distributions. I didn’t see anywhere in the analyses where scattering depth was considered. It would be very interesting to know if depth distributions changed with proximity to platforms.

6. PLOS authors have the option to publish the peer review history of their article (what does this mean?). If published, this will include your full peer review and any attached files.

Reviewer #1: No

Reviewer #2: No

---

## [Author Response · Author response to Decision Letter 0]

15 Mar 2024

Responses to all editor and reviewer comments are included in the updated cover letter included with this resubmission. The following is copied directly from that letter:

Responses to reviewer comments (review comments in black, our response in red, note: lines given denote position in the marked-up version of the manuscript, not the clean version, for better clarity on what edits were made)

Reviewer #1: 

Line 158: corrected depth please clarify

The term ‘corrected’ has been removed for clarity – this simply referred to the process of checking and correcting the automatically detected seabed line as part of the pre-processing of the acoustic data (also edited line 123 to clarify this step)

Line 187: please name the ICES data portal

This has been added (see lines 211)

Page 13: I would prefer to include the Figure and Table from the Supplementary material here in the main MS

Figure and table moved to the main body.

Line 283: visual inspection of the variograms or formal test?

The variograms were inspected visually; text has been edited for clarity. (Line 330)

Line 333: what about the dead zone at the surface? Was a drop keel used?

This paragraph is talking about the limitations on detecting fish around pipelines on the seabed where the bottom dead-zone is relevant, but the surface dead-zone is not. It is, however, mentioned that the data at ranges between 0-12m from the transducer face were excluded (as the near field) from analyses as part of pre-processing of acoustic data. (Line 122)

Line 355: Sediment maps for the North Sea should be available from geological surveys or projects

We have added text and a citation to mention these habitat maps (line 452). However, while these maps are available, in the past some authors have found issues with their fine-scale accuracy and reliability in the areas where they rely on interpolation. Ideally, additional work involving acoustic characterisation, drop-camera/grab sampling and comparison with existing seabed habitat maps would, hopefully, allow the reconciliation of these complementary datasets and the reduction of uncertainty around seabed characterisation. 

Page 18: Please mention that the results and conclusions may warrant a validity check with an additional data set, e.g. from quarter 1 survey or another year or a special study with bottom and pelagic trawl catches in addition to the acoustics

Text has been added to the discussion to address these suggestions. (Line 477 and 423)

 

Reviewer #2: 

Overall a nice paper. The authors have done considerable work to address fish distribution and proximity to platforms. I think the manuscript could be strengthened by the addition of a couple of analyses that should not be too arduous to complete. I provide more detail in general comment #4 and specific comment #22. My recommendation is major revisions.

General Comments:

Lines 17-19. I find this statement interesting. Is the assumption that the authors are referring to the pelagic environment, rather than the benthic and/or demersal environments? If so, I don’t quite agree that the pelagic is featureless (and I don’t understand the use of “relatively”. Relative to what?). I think there are features, but we as humans either don’t understand them and consequently don’t measure them, or choose to ignore them. If the benthic habitat is to be considered, then I really don’t agree with it being featureless. I think this statement needs more detail and clarification.

Re-worded for clarity and specificity (line 20)

An echogram would be very useful to give perspective and context.

Added an echogram showing fish schools and aggregation of individual fish in proximity to an oil platform as an illustrative example.

I wasn’t sure how day/night fit into the analyses. The results state day/night differences, but were the data separated into day & night prior to applying statistics or was day/night a covariate? For example, Figure S1 shows only one trend line per graph, suggesting the data were pooled day & night, but if there were day/night differences, shouldn’t there be two graphs per test?

Day/night was included as a covariate, but as a factor variable (mentioned lines 280-2), so no additional plots of smooth terms were generated. Instead, a term is included in the model which adds an additional effect during ‘day’ as opposed to ‘night’, which is included in the model intercept. Additional tables of results for factor terms have been added to the supplementary materials.

Was it possible to calculate fish length from the TS measurements? It would be very interesting to see if there were any relationships between fish length and proximity to platforms.

We agree that this is an interesting and important question, and one we are working on as part of other pieces of work, but we consider it beyond the scope of this study. The calculation of fish lengths from the TS measurements relies entirely upon the correct assumption of species composition (or more precisely, the selection of appropriate TS-L relationships). While trawl data was collected during the survey, it was not targeted/designed to inform the acoustic data, and so would introduce varying uncertainty in species assignation due to varying degrees of spatial mismatch between the acoustic data and trawl datasets. Additionally, the comment about unsampled pelagic scattering is relevant here, and adds additional uncertainty in the selection of appropriate TS-L relationships. However, as we agree this is of interest, a paragraph has been added to the discussion raising this as an important future question, as well as the ‘specific comment’ about potential for TS bias from lack of pelagic sampling. (Lines 479-487 & 415-426)

Specific Comments:

Line 20. There is still considerable debate about “spill over”. I would temper this statement with “potentially”.

Added ‘potentially’

Line 79. Delete “very”.

Deleted

Line 90. Please define what “the region” refers to. The authors have referred to a number of different regions around the globe. Maybe rewrite to “... platforms throughout the North Sea.” if they are only referring to the North Sea.

Edited as suggested (line 93)

Line 106. Please provide the beam widths. This is important information for some of the processing steps.

Added (line 110)

Line 110. Should the comma after “performed” be a colon?

Changed

Line 112. Was there a backstep applied to the seabed echo detection?

Added (123)

Lines 123-124. Is there a reference for “harder edges”?

Text edited slightly for clarity, and references added (line 136-7)

Lines 124-126. Please state how the “area around each candidate school” was selected.

Text added (line 138)

Line 139. Please state which dimension each value refers to, e.g., 50 m horizontal by 5 m vertical.

Added (line 153)

Line 141. Please define “masked”.

Text added (line 155)

Line 143. What type of trawl was used? If a bottom trawl, do the authors suspect a bias in the TS calculations for pelagic scattering? For example, juvenile gadoids are often in the water column, whereas adults are demersal. Using a TS for adults would not be representative of the individuals in the water column.

It was a demersal trawl (GOV trawl, BT137) that was used, and so there may be a bias in the TS estimation due to lack of pelagic sampling. Large numbers of small fish (e.g. below 7cm) were caught during the survey, so the bottom trawl does not exclude fish of that size, but it is true that there will be a bias based on the non-random distribution of the size range of fish species of interest here. Comment has been made in the discussion about this. (Line 415-426)

Lines 166-168. I recommend the authors look at Kieser and Mulligan, 1984, Analysis of echo counting data: A model, Canadian Journal of Fisheries and Aquatic Sciences, 41: 451-458 as a method to estimate echo count density.

This is an interesting reference which we have added a citation for (line 190), and we believe it supports our methods – equation (2) in Kieser & Mulligan is essentially the same as our equation (3) on line 187 (DensEDSU = …), except we account for ping-wise variations in the depth, and we calculate fish density per unit area (using depth/volume), whereas Kieser & Mulligan calculate fish density per unit volume (using 1/volume).

Line 187. A URL to the ICES data portal would be useful.

Added, line 214

Line 191. What were the criteria for day and night from sun elevation?

Text added to clarify (line 218)

Line 204. “Data” is plural. Replace “was” with “were”.

Corrected

Line 229. “Gaussian” should be capitalized.

Corrected

Line 238. Specify which analyses. In the previous section, data >25 km were used for the baseline.

Clarified (line 283)

Table 1 vs. Figure S1. I like the figure over the table. I feel the figure is more informative than the table, so I suggest replacing Table 1 with Figure S1.

Figure S1 moved to the main body, to replace Table 1.

Paragraph beginning on line 270. I would like to see a figure representing the spatial autocorrelation. This could be in the supporting documentation.

A figure showing residual variogram and autocorrelation for all models added to supporting information.

Line 292. The validity of this paragraph is impossible to evaluate without any statistical test, results, or figures. Either provide details or delete.

Additional tables of results for factor variables have been added to the supplementary materials to show the data needed.

The discussion should include a paragraph on seasonality and migration behaviour. It would be interesting to know how their survey timing fits into the overall distribution of the key species.

The discussion should also cover pelagic vs. demersal distributions. I didn’t see anywhere in the analyses where scattering depth was considered. It would be very interesting to know if depth distributions changed with proximity to platforms.

Paragraphs have been added to the discussion to discuss seasonality, migration and spawning timing/activity (lines 460-478). The discussion now also includes mention of the potential for change in vertical distribution of demersal fish (lines 427-443), although we felt this analysis was outside the scope of the current work, which focussed purely on the horizontal trends in distribution. Future work is planned to examine evidence for changes in vertical distribution in proximity to these platforms.

---

## [Editor Report · Decision Letter 1]

11 Apr 2024

Elevated fish densities extend kilometres from oil and gas platforms

PONE-D-23-35757R1

Dear Dr. Lawrence,

We’re pleased to inform you that your manuscript has been judged scientifically suitable for publication and will be formally accepted for publication once it meets all outstanding technical requirements.

Kind regards,

Brian R. MacKenzie, Ph. D.

Academic Editor

PLOS ONE